# READY-TO-REACT: ONLINE REACTION POLICY FOR TWO-CHARACTER INTERACTION GENERATION

**Zhi Cen**[1], **Huaijin Pi**[2], **Sida Peng**[1], **Qing Shuai**[1], **Yujun Shen**[3], **Hujun Bao**[1], **Xiaowei Zhou**[1], **Ruizhen Hu**[4*]

[1]State Key Lab of CAD&CG, Zhejiang University, [2]The University of Hong Kong, [3]Ant Group, [4]Shenzhen University

zhicen@zju.edu.cn, ruizhen.hu@gmail.com

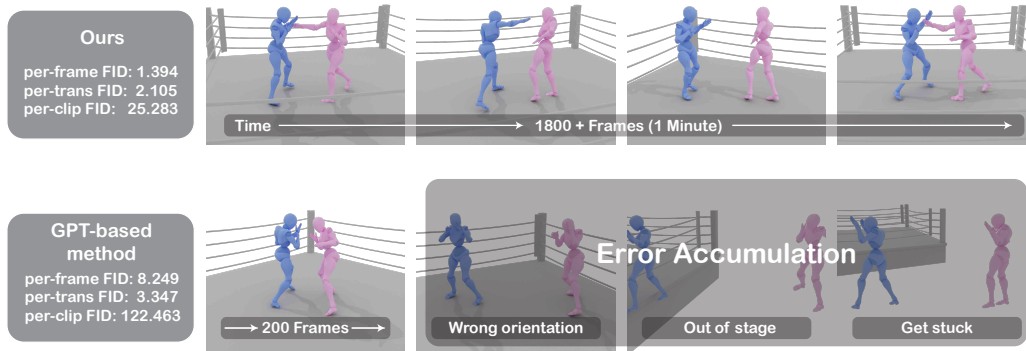

Figure 1: **Demonstration of Ready-to-React**, an *online* reaction policy for two-character interaction generation on the challenging task of boxing. Ready-to-React predicts the next pose of an agent by considering its own and the counterpart's historical motions. Our method can successfully generate 1800 frames of motion, whereas the GPT-based approach struggles after about 200 frames, displaying issues such as incorrect orientation, leaving the ring boundary, or freezing in place due to the accumulation of errors over time.

## ABSTRACT

This paper addresses the task of generating two-character online interactions. Previously, two main settings existed for two-character interaction generation: (1) generating one's motions based on the counterpart's complete motion sequence, and (2) jointly generating two-character motions based on specific conditions. We argue that these settings fail to model the process of real-life two-character interactions, where humans will react to their counterparts in real time and act as independent individuals. In contrast, we propose an online reaction policy, called Ready-to-React, to generate the next character pose based on past observed motions. Each character has its own reaction policy as its "brain", enabling them to interact like real humans in a streaming manner. Our policy is implemented by incorporating a diffusion head into an auto-regressive model, which can dynamically respond to the counterpart's motions while effectively mitigating the error accumulation throughout the generation process. We conduct comprehensive experiments using the challenging boxing task. Experimental results demonstrate that our method outperforms existing baselines and can generate extended motion sequences. Additionally, we show that our approach can be controlled by sparse signals, making it well-suited for VR and other online interactive environments. Code and data will be made publicly available at https://zju3dv.github.io/ready_to_react/.

## 1 INTRODUCTION

This paper aims to learn a reaction policy from data that can generate two-character interactions in a streaming manner. Such a policy is essential for applications in robotics, gaming, and virtual

---

*Corresponding author.

reality, where the character needs to interact with other entities in real-time. Generating reasonable online reactions is quite challenging, considering two key perspectives. Firstly, the policy should dynamically adjust its own actions based on the counterpart's responses at each time step, which is essential for enabling online applications. Secondly, downstream applications require the policy to generate natural and physically plausible motions while maintaining consistency and diversity throughout these sequences.

Previously, there were two main settings to generate two-character interaction: (1) generating one's motions based on a complete sequence of the counterpart's motion (Siyao et al., 2024; Ghosh et al., 2024; Xu et al., 2024b), and (2) jointly generating two-character motion sequences based on specific conditions, such as textual input (Tanaka & Fujiwara, 2023; Gu et al., 2024; Ruiz-Ponce et al., 2024). However, we argue that both settings do not model the process of real-life two-person interactions. As humans, we dynamically produce the reaction based on the counterpart's actions at each time step, engaging in independent mind rather than sharing a collective consciousness. To achieve two-character online interaction, it is crucial to develop a reaction policy that can be separately applied to two characters, allowing them to react to each other like real humans.

In this paper, we propose a novel reaction policy, called Ready-to-React, for generating two-character interactions, as illustrated in Figure 2. Our core innovation lies in incorporating a diffusion head into an auto-regressive model, which can respond to the counterpart's motions in a streaming manner while ensuring the naturalness and diversity of the motions. Specifically, we first encode the observed motions into latent vectors using a motion encoder. Given the history motion latent of the character and the counterpart's motion, our reaction policy uses an auto-regressive model to predict a conditioning feature vector. This vector guides a diffusion model to generate the next motion latent, which is then decoded into the next character pose. As illustrated in Figure 1, the proposed reaction policy can generate long and natural boxing sequences compared to the baseline method.

We chose boxing as the task for online two-character interaction generation considering its fast pace and frequent shifts between offense and defense. These dynamic interactions make it an ideal scenario for testing and validating our approach. We conducted experiments to validate the effectiveness of our approach on our self-collected boxing dataset DuoBox. Our method was evaluated under three settings: (1) reactive motion generation, (2) two-character interaction generation, and (3) long-term two-character interaction generation, where it outperforms the baselines. We show that our method can generate very long motion sequences ($\sim$ 1 minute), just relying on the initial poses of the two characters. Additionally, we carried out experiments on sparse control motion generation, demonstrating that our method is well-suited for VR online interactive settings.

## 2 RELATED WORK

**Single-character motion generation.** In recent years, deep learning methods for motion synthesis have gained increasing attention (Fragkiadaki et al., 2015; Holden et al., 2016; Martinez et al., 2017; Dou et al., 2023; Xie et al., 2023; Pi et al., 2023; Cen et al., 2024). Various techniques, including multi-layer perceptron (MLP) (Holden et al., 2017), mixture of experts (MoE) (Zhang et al., 2018), and recurrent neural networks (RNN) (Harvey et al., 2020) are employed to tackle this task. Additionally, to generate diverse and natural results, generative models such as conditional variational auto-encoders (cVAE) (Ling et al., 2020), generative adversarial networks (GAN) (Li et al., 2022), and normalizing flows (Henter et al., 2020) have shown promise in addressing this challenge. Moreover, the success of generative pre-trained transformers (GPT) (Zhang et al., 2023) and diffusion models (Tevet et al., 2023) further underscores their potential in this area. Recently, Chen et al. (2024); Shi et al. (2024) adopt auto-regressive diffusion models to generate single-character motions. Although our reaction policy generates one character's motion based on the other's, technically making it single-character motion generation, our work focuses on decision independence in two-character interactions.

**Reactive motion generation.** Reactive motion generation is a subfield of human motion generation that focuses on generating human motion in response to external agents. Men et al. (2022) introduces a semi-supervised GAN system with a part-based LSTM module to model temporal significance. Chopin et al. (2023) employs a transformer network, enhanced by an interaction distance module using graphs. Siyao et al. (2024) proposes a GPT-based model to predict discrete motion

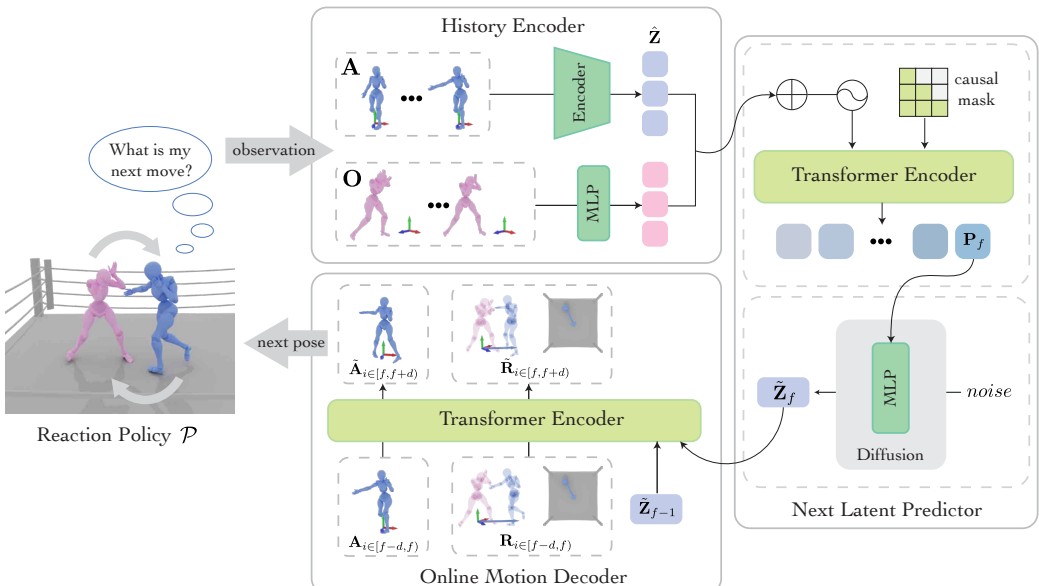

Figure 2: **Pipeline overview.** Given a boxing scene at the leftmost figure, where the blue agent is thinking about its next move. The reaction policy (Section 3.2) follows these steps: first, based on the observations, the history encoder encodes the current state and observations; then, the next latent predictor predicts the upcoming motion latent; and finally, an online motion decoder decodes this motion latent into the actual next pose. The same reaction policy can be applied to the pink agent. Through a streaming process for both agents, our reaction policy enables the continuous generation of two-character motion sequences without length limit (Section 3.3).

tokens, enhanced by an off-policy reinforcement learning strategy. Xu et al. (2024b) utilizes a diffusion-based model with an explicit distance-based interaction loss. Ghosh et al. (2024) employs a diffusion-based model with a combined spatio-temporal cross-attention mechanism. However, Xu et al. (2024b) and Ghosh et al. (2024) require the complete motion sequence of the other agent, and cannot generate reactive motions online. In contrast, our method enables online interaction generation by leveraging auto-regressive diffusion models, as inspired by Li et al. (2024).

**Two-character motion generation.** While multi-character motion generation focuses on producing motion for groups with social relationships (Zhu et al., 2023b; Peng et al., 2023; Le et al., 2023; Lim et al., 2023; Jeong et al., 2024), two-character motion generation focuses on generating closer interaction between two characters. Starke et al. (2020) proposes the local motion phase for complex, contact-rich interactions. Starke et al. (2021) combines a motion generator with task-dependent control modules. Both Starke et al. (2020) and Starke et al. (2021) require user control signals. Physically-based methods (Won et al., 2021; Zhu et al., 2023a; Younes et al., 2023) ensure the physical plausibility of character interactions but still struggle to generate natural and diverse motions. Liu et al. (2019); Fieraru et al. (2020); Yin et al. (2023); Liang et al. (2024); Xu et al. (2024a) provide datasets with rich annotations that enable advancements in interaction modeling two-character scenarios. Recent advances in text-driven two-character motion generation have introduced diffusion models (Tevet et al., 2023) to enhance realism and control (Tanaka & Fujiwara, 2023; Gu et al., 2024; Ruiz-Ponce et al., 2024; Cai et al., 2024). Despite these advancements, these methods jointly generate whole sequences for both characters and ignore the dynamic feedback inherent in real-life two-person interactions.

## 3 METHOD

Our aim is to develop an intelligent reaction policy capable of generating online motions that dynamically respond to a counterpart's movements. We begin by outlining the formulation of the one-step reaction in Section 3.1. Then, we introduce the design of the reaction policy in Section 3.2. We ex-

plain how this policy drives the generation of two-character motions in Section 3.3. In Section 3.4, we explain the training loss and implementation details.

## 3.1 PROBLEM FORMULATION

We first introduce the problem formulation. There are two characters, the agent $\mathcal{A}$ and the opponent $\mathcal{O}$. The goal is to generate the future agent's motion $\mathbf{A}_f$ based on the opponent's movements $\mathbf{O}_{i \in [f-W, f)}$ and the agent's previous motion $\mathbf{A}_{i \in [f-W, f)}$:

$$\mathbf{A}_f = \mathcal{P}(\mathbf{O}_{i \in [f-W,f)}, \ \mathbf{A}_{i \in [f-W,f)}), \tag{1}$$

where $\mathcal{P}$ is the generative reaction policy, and $W$ is the visible past window size. We use the same root coordinate definition in Starke et al. (2020; 2021), as shown in Figure 2. Each agent's motion $\mathbf{A}_i$ is defined as:

$$\mathbf{A}_i = \{\mathbf{r}_{\text{off}}^{\mathcal{A}} \in \mathbb{R}^2, \ \mathbf{r}_{\text{dir}}^{\mathcal{A}} \in \mathbb{R}^2, \ \Theta_{\text{pos}}^{\mathcal{A}} \in \mathbb{R}^{J \times 3}, \ \Theta_{\text{rot}}^{\mathcal{A}} \in \mathbb{R}^{J \times 6}, \ \Theta_{\text{vel}}^{\mathcal{A}} \in \mathbb{R}^{J \times 3}\}, \tag{2}$$

where $\mathbf{r}_{\text{off}}^{\mathcal{A}}$ and $\mathbf{r}_{\text{dir}}^{\mathcal{A}}$ are the horizontal trajectory (excluding the y-axis) positions and directions relative to the $(i-1)$-th frame agent's root coordinate, respectively. The $\Theta_{\text{pos}}^{\mathcal{A}}$, $\Theta_{\text{rot}}^{\mathcal{A}}$ and $\Theta_{\text{vel}}^{\mathcal{A}}$ are the positions, 6D rotations and velocities relative to the $i$-th frame agent's root coordinate, respectively. $J$ is the number of body joints. Each opponent's motion $\mathbf{O}_i$ is defined as:

$$\mathbf{O}_i = \{\ddot{\Theta}_{\text{pos}}^{\mathcal{A}} \in \mathbb{R}^{J \times 3}, \ \ddot{\Theta}_{\text{rot}}^{\mathcal{A}} \in \mathbb{R}^{J \times 6}, \ \ddot{\Theta}_{\text{vel}}^{\mathcal{A}} \in \mathbb{R}^{J \times 3}\}, \tag{3}$$

which includes the positions, 6D rotations and velocities relative to the $i$-th frame agent's ($\mathcal{A}$'s) root coordinate, respectively.

## 3.2 REACTION POLICY

In this section, we present our reaction policy, which predicts motion in the latent space. As illustrated in Figure 2, the history encoder module is responsible for encoding the observations of the current state. Based on this historical information and the opponent's motion, the next latent predictor module predicts the future motion latent codes. Finally, the online motion decoder generates the future motions using the predicted latent and the current agent's state.

**Latent motion representation.** Converting raw data to latent space has become a popular approach in generation pipelines (Rombach et al., 2022), and VQ-VAE has been proven to be effective in learning disentangled representations of human motion data (Zhang et al., 2023; Jiang et al., 2024; Lee et al., 2024; Starke et al., 2024). In this paper, we adopt a similar architecture as in Zhang et al. (2023) to encode the raw motion data into latent sequences. Given a sequence of agent motion $\mathbf{A} = \{\mathbf{A}_i \mid i \in [0, f), \ i \in \mathbb{Z}\}$, the VQ-VAE motion encoder encodes it to $\mathbf{Z} = \{\mathbf{Z}_i \mid i \in [0, \lfloor \frac{f}{d} \rfloor), \ i \in \mathbb{Z}\}$ with a downsampling factor $d = 4$, and each latent code $\mathbf{Z}_i$ is quantized through the codebook $\mathcal{C}$ to find the most similar element:

$$\hat{\mathbf{Z}}_i = \arg\min_{\mathcal{C}_k \in \mathcal{C}} \|\mathbf{Z}_i - \mathcal{C}_k\|_2. \tag{4}$$

**History encoder.** First, we need to encode the past information of both the agent and the opponent. To represent the agent's past motion $\mathbf{A}$, we could directly utilize the VQ-VAE motion encoder to compress them into latent variables $\hat{\mathbf{Z}}$. The opponent's historical motion $\mathbf{O}$ is also downsampled by a factor of $d = 4$ and then passed through a single-layer MLP, which encodes it into a feature vector with the dimension of $\hat{\mathbf{Z}}$, ensuring consistency for subsequent processing.

**Next latent predictor.** We then predict the next motion latent in an auto-regressive manner based on the historical information. Our approach employs a transformer-based condition encoder to effectively capture all visible information. This encoded data is then passed to the diffusion-based motion latent predictor, which generates the next motion latent based on the encoded conditions.

Specifically, we begin by constructing a transformer-based encoder to encode the information accessible to the reaction policy. As illustrated in Figure 2, the transformer's input consists of the motion latent code $\hat{\mathbf{Z}}$ and the opponent's feature obtained from the history encoder. As a result, each output of the transformer-based encoder $\mathbf{P}_f$ encapsulates the information preceding the $f$-th frame.

Next, we introduce the diffusion process, which predicts the future motion latent codes $\tilde{\mathbf{Z}}_f$ based on $\mathbf{P}_f$ at the $f$-th frame. We use conditional diffusion models (Tevet et al., 2023; Ramesh et al., 2022) and $\mathbf{P}_f$ serves as the conditioning input, as shown in Figure 2. We employ a single-layer MLP as the generative model $\mathcal{G}$, ensuring that our model can operate in real-time. The predicted motion latent code $\tilde{\mathbf{Z}}_f$ is then used to generate the future motion through the online motion decoder.

Compared to previous methods that rely on predicting motion tokens' probabilities and supervising GPT models with cross-entropy loss, our approach of predicting motion latent using a diffusion-based model preserves the smooth and continuous nature of motion. This results in fewer cumulative errors and less deviation from the intended motion over time, offering greater stability and accuracy than predicting token probabilities with GPT models.

**Online motion decoder.** A remaining problem is to decode the predicted motion latent code $\tilde{\mathbf{Z}}_f$ into the future agent motion $\tilde{\mathbf{A}}_{i \in [f, f+d)}$ online. We propose an online motion decoder that takes a few previous frames and two consecutive motion latent codes to generate the next motion frames. As shown in Figure 2, we use a transformer as the online motion decoder. The inputs to the online motion decoder are the past agent motions $\mathbf{A}_{i \in [f-d, f)}$, root information $\mathbf{R}_{i \in [f-d, f)}$, the last motion latent code $\tilde{\mathbf{Z}}_{f-1}$ and current motion latent code $\tilde{\mathbf{Z}}_f$ from the next latent predictor. Here, each root information $\mathbf{R}_i$ is defined as:

$$\mathbf{R}_i = \{\mathbf{r}_{\text{off}}^{\mathcal{O}} \in \mathbb{R}^2, \ \mathbf{r}_{\text{dir}}^{\mathcal{O}} \in \mathbb{R}^2, \ \mathbf{r}_{\text{dis}} \in \mathbb{R}^1\}, \tag{5}$$

which includes the agent's horizontal trajectory (excluding the y-axis) positions and directions relative to the opponent's root coordinate, and the distance between the agent's root and the center of the boxing ring. The output of the online motion decoder are the future agent motions $\tilde{\mathbf{A}}_{i \in [f, f+d)}$ and the future root information $\tilde{\mathbf{R}}_{i \in [f, f+d)}$. In contrast to VQ-VAE decoder (Zhang et al., 2023), which requires the entire sequence of tokens before decoding, our method decodes motion latent into explicit motion sequences in real-time using only a few tokens and historical data, enabling online generation.

### 3.3 ONLINE TWO-CHARACTER MOTION GENERATION

Our reaction policy, as described in Section 3.2, enables the generation of the next motion frame by leveraging both the opponent's past motion and the agent's own past motion. To implement online two-character interaction generation, we use the same reaction policy $\mathcal{P}$ for the two characters.

Starting with an initial input of $s = 4$ frames of poses, both characters use the policy $\mathcal{P}$ to generate the next $d$ frames by considering their own initial motion and the opponent's motion. These generated motions are then added to a history buffer $\mathcal{H}$ with a maximum size of $W$. After this initial phase, both characters continuously update their predictions by incorporating the newly generated frames and the accumulated motion history. The interaction generation process operates in a streaming fashion. Motion that exceeds the buffer size $W$ is discarded as outdated information. Through this approach, both characters dynamically respond to the other, ensuring coherent and natural interactions while enabling two-character motion generation without a length limit.

### 3.4 TRAINING LOSS AND IMPLEMENTATION DETAILS

The training process is divided into two stages: (1) pre-training the VQ-VAE model and (2) jointly training the next latent predictor model and the online motion decoder. Details about the network architecture are provided in Appendix C.

**Stage 1.** We pre-train the VQ-VAE model following the approach in Zhang et al. (2023); Razavi et al. (2019) for $40k$ iterations, using motion sequences cropped to 64 frames. The batch size is set to 128, with a codebook size = 512, codebook feature dimension = 512, and a downsampling rate of $d = 4$. The codebook is updated with the exponential moving average (EMA) method (Razavi et al., 2019), as a replacement for the codebook loss. Finally, the loss is defined as:

$$\mathcal{L}_{\text{vqvae}} = \mathcal{L}_{rec} + \alpha \|sg[\hat{\mathbf{Z}}] - \mathbf{Z}\|_2^2, \tag{6}$$

where $\mathcal{L}_{rec}$ is the L2 reconstruction loss, $\|sg[\hat{\mathbf{Z}}] - \mathbf{Z}\|_2^2$ is the commitment loss, the operator $sg$ refers to a stop-gradient operation, and $\alpha = 0.1$.

**Stage 2.** Next, we train the next latent predictor and online motion decoder jointly for $40k$ iterations while keeping the VQ-VAE motion encoder fixed. During training, we applied causal masks (Figure 2) to ensure that the model can only access the current and previous inputs, preventing information from leaking into future time steps. We use ground truth $\hat{\mathbf{Z}}_{f-1}$ instead of the predicted $\tilde{\mathbf{Z}}_{f-1}$ in Figure 2 during training. For this phase, motion sequences are cropped to $W = 60$ frames (2 seconds) for training. The batch size is set to 32, with time step $T = 1000$, and we employ DDIM (Song et al., 2020) to sample only 50 steps during inference. The loss is defined as:

$$\mathcal{L} = \mathcal{L}_{\text{diffusion}} + \beta\|\mathbf{A} - \tilde{\mathbf{A}}\|_2^2 + \gamma\|\mathbf{R} - \tilde{\mathbf{R}}\|_2^2, \tag{7}$$

where $\beta = 1.0$, $\gamma = 1.0$. $\mathcal{L}_{\text{diffusion}}$ is defined as:

$$\mathcal{L}_{\text{diffusion}} = \mathrm{E}_{t\in[1,T],\mathbf{x}_0\sim q(\mathbf{x}_0)}\left[\|\mathbf{x}_0 - \mathcal{G}(\mathbf{x}_t, t, \mathbf{c})\|_2\right], \tag{8}$$

where $\mathbf{x}_0 = \hat{\mathbf{Z}}$ is the ground truth next latent, $\mathcal{G}$ is the generative model, $\mathbf{c} = \mathbf{P}_f$ is the condition, and $\mathcal{G}(\mathbf{x}_t, t, \mathbf{c}) = \tilde{\mathbf{Z}}$ is the predicted next latent. All models are trained using the AdamW optimizer (Kingma & Ba, 2014) with a learning rate of 0.0001 on a single Nvidia RTX 4090 GPU.

## 4 EXPERIMENTS

### 4.1 DATASET, EXPERIMENTAL SETTING, AND EVALUATION METRICS

**Dataset.** To evaluate our method, we collect a high-quality dataset, DuoBox, using the OptiTrack Mocap system[1] equipped with 12 cameras. We invited three boxing enthusiasts to perform various boxing movements, recording multiple sequences of their actions. Please refer to Appendix A for the details of the data collection. In total, DuoBox consists of 63.4 minutes of motion data (approximately 457k frames) captured at 120 FPS. For our experiments, we split the dataset into training (80%) and testing (20%) subsets, and downsample the original data to 30 FPS for training purposes. To further enrich the dataset, we apply the augmentation by swapping the roles of the agent and the opponent during training.

**Experimental setting.** We evaluate our method in three scenarios: reactive motion generation, two-character interaction generation, and long-term two-character interaction generation. In the reactive motion generation, we use the ground truth for the opponent's motion as input. For the two-character interaction generation, we provide only the initial 4 frames of poses for both characters. In both test scenarios, the motion of individual characters follows the procedure outlined in Section 3.2, while the generation of two-character motions adheres to the process described in Section 3.3.

**Evaluation metrics.** We evaluate the generated motion sequences using the following metrics: (1) **Frechet Inception Distance (FID).** We follow Dou et al. (2023) calculating per-frame, per-transition and per-clip FIDs. A lower FID indicates the generated motion is more similar to the real data. (2) **Jitter.** We evaluate motion jittery following Shen et al. (2024). A closer value to the ground truth indicates better motion quality. (3) **Root Orient (RO).** To assess the long-term consistency of the generated motion, we propose a new metric RO. It calculates the percentage of frames where the facing direction between the two agents exceeds 45 degrees. This is motivated by the nature of boxing, where the athletes typically face to each other throughout the match. Lower deviation from the ground truth means the generated motion aligns better with the expected interactive behavior. (4) **Foot Sliding (FS).** Foot sliding (Ling et al., 2020) measures the average sliding distance when the foot is close to the ground ($< 5cm$). A value closer to the ground truth indicates better motion quality. Minimal foot slide may suggest that the generated motion remains stationary. In addition, we provide the inference speed in Appendix E and motion diversity analysis in Appendix F.

### 4.2 COMPARISON WITH BASELINES

We compare our method against several baselines, including InterFormer (Chopin et al., 2023), CVAE-AR, CAMDM (Chen et al., 2024), T2MGPT-online (Zhang et al., 2023), and Duolando-offline (Siyao et al., 2024). Among these, InterFormer is a deterministic model and lacks the ability to generate diverse results. CVAE-AR is a CVAE-based approach that we construct specifically for

---

[1]https://optitrack.com/

Table 1: **Comparison with baselines.** We compare our method with five baselines (Section 4.2) in the two scenarios: reactive motion generation and two-character motion generation. Among them, **bold** indicates the best results. ↓ means lower is better. → means closer to the real data is better.

| Methods | Reactive | | | | | | Two-character | | | | | |
| | FID↓ | | | Jitter→ | RO→ | FS→ | FID↓ | | | Jitter→ | RO→ | FS→ |
| | Per-frame | Per-trans. | Per-clip | | | | Per-frame | Per-trans. | Per-clip | | | |
| Real | - | - | - | 21.332 | 24.7% | 0.97 | - | - | - | 21.332 | 24.7% | 0.97 |
| InterFormer | 0.724 | 1.993 | 15.061 | 6.712 | 40.3% | 1.08 | 2.498 | 4.590 | 47.194 | 6.117 | 31.5% | 1.05 |
| CVAE-AR | 1.285 | 4.233 | 26.010 | 28.519 | 46.2% | 1.13 | 5.405 | 9.006 | 92.978 | 28.037 | 38.4% | 1.09 |
| CAMDM | 1.606 | 4.037 | 28.503 | 53.622 | 40.1% | 1.96 | 4.162 | 7.488 | 70.416 | 53.994 | 22.6% | 2.01 |
| T2MGPT-online | 1.721 | 1.567 | 30.159 | 78.292 | 50.4% | 2.65 | 8.249 | 3.347 | 122.463 | 78.334 | 46.2% | 2.63 |
| Duolando-offline | 1.025 | 5.862 | 13.606 | **22.544** | **30.8%** | 3.19 | - | - | - | - | - | - |
| Ours | **0.535** | **0.995** | **9.599** | 17.825 | 34.7% | **1.02** | **1.394** | **2.105** | **25.283** | **16.844** | **24.1%** | **0.97** |

Table 2: **Quantitative results of long-term two-character motion generation.** We compare our method with four baselines (Section 4.2). The generated motion lengths are set to 1800 frames.

| Methods | FID ↓ | | | Jitter → | RO → | FS → |
| | Per-frame | Per-transition | Per-clip | | | |
| Real | - | - | - | 21.332 | 24.7% | 0.97 |
| InterFormer | 5.628 | 5.936 | 87.697 | 4.784 | 44.6% | **0.90** |
| CVAE-AR | 7.325 | 11.717 | 110.458 | 18.107 | 36.7% | 0.75 |
| CAMDM | 7.557 | 13.465 | 109.654 | 25.237 | 13.7% | 0.86 |
| T2MGPT-online | 21.27 | 4.457 | 273.086 | 72.481 | 70.4% | 2.34 |
| Ours | **2.388** | **2.375** | **36.755** | 19.204 | **31.3%** | 0.83 |

comparison. For CAMDM, we modify its input to make it function as a reaction policy. T2MGPT-online refers to decoding each newly generated token immediately into raw motion using a VQ-VAE decoder as soon as the token is produced. Duolando-offline (Siyao et al., 2024) decodes the tokens after the entire sequence has been generated. We retrain all these methods on the DuoBox using similar training configurations for a fair comparison. Details can be found in Appendix D.

**Reactive motion generation.** We begin by evaluating our method in the context of generating reactive motion, where the opponent's motion is provided as ground truth. The results, presented in Table 1 (left: reactive), show that our method significantly outperforms the baseline across multiple metrics, including per-frame, per-transition, and per-clip FID scores, as well as reducing foot sliding. Additionally, the RO of our method closely matches the performance of the offline Duolando generation. Notably, the root prediction in Duolando is relative to the opponent, preventing drift over time. Among the online prediction methods, InterFormer may generate motions with implausible root position and orientation. T2MGPT, which decodes GPT-predicted tokens online, exhibits jitter and discontinuity. Both CVAE-AR and CAMDM produce motion that is easy to get stuck over time. We also provide qualitative results in Figure 5 and the supplementary materials.

**Two-character interaction generation.** Our method enables the simultaneous generation of motion for both agents. Starting with the first four frames, each agent's subsequent motion is generated by leveraging the interaction between their own and their opponent's past motions. In contrast, Duolando-offline cannot generate both agents' motions simultaneously, as it predicts the root position relative to the opponent's at the same frame and requires future information through a looking-ahead mechanism. Moreover, it relies on having all tokens available before decoding, preventing online generation. As shown in Table 1 (right: two-character), our method significantly outperforms the baseline across all metrics in this more complex scenario. We also provide qualitative results in Figure 6 and the supplementary materials.

**Long-term two-character interaction generation.** We demonstrate our method's ability to generate extended sequences of two-character motion, highlighting its reduced error accumulation and superior motion quality over long durations. Our method is capable of generating two-character motion for 1800 frames. The results are shown in Table 2. The FID of ours is much better than all the baselines. To show the effect of error accumulation, we plot the face direction relative to time in Figure 3. T2MGPT-online and InterFormer will quickly generate motions that face away from each other after some time, and CAMDM generates motions that are over-smoothed. Please refer to the supplementary material for more visualizations.

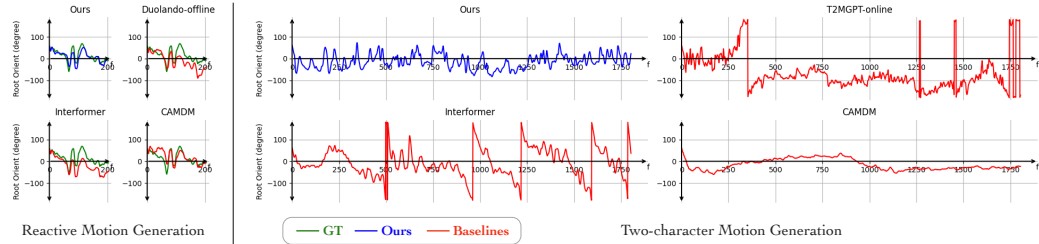

Figure 3: **Visualization of the face direction relative to time.** We compare our method with baselines in two scenarios described in Section 4.2. The x-axis represents the frame number f, while the y-axis shows the angle between the facing directions of the two characters (in degrees). An angle of $0°$ indicates that the two agents are facing each other, whereas $\pm 180°$ means they are facing away from each other. The green lines represent the ground truth, the blue lines represent our method, and the red lines represent the baselines.

Table 3: **Ablation study.** We compare our method with five variants to validate our main design choices (please refer to Section 4.3 for details). Among them, **bold** indicates the best results. ↓ means lower is better. → means closer to the real data is better.

| Methods | Reactive | | | | | Two-character | | | | |
|---|---|---|---|---|---|---|---|---|---|---|
| | FID↓ | | | RO→ | FS→ | FID↓ | | | RO→ | FS→ |
| | Per-frame | Per-trans. | Per-clip | | | Per-frame | Per-trans. | Per-clip | | |
| Real | - | - | - | 24.7% | 0.97 | - | - | - | 24.7% | 0.97 |
| use VAE as encoder | 0.525 | 1.290 | 10.423 | 36.5% | 1.00 | 1.446 | 3.002 | 28.268 | 22.2% | 1.01 |
| w/o motion encoder | 0.693 | 1.339 | 24.860 | 54.0% | 1.48 | 7.203 | 3.012 | 126.991 | 52.3% | 1.47 |
| use GPT | 0.892 | 1.781 | 16.212 | 37.4% | **0.99** | 2.418 | 3.536 | 41.144 | 20.9% | **0.97** |
| w/o online decoder | 5.215 | 11.020 | 78.933 | 44.9% | 1.06 | 11.279 | 23.189 | 157.253 | 16.0% | 0.91 |
| w/o **R** in decoder | **0.496** | 1.074 | 10.315 | 43.3% | 0.93 | 3.370 | 2.275 | 54.503 | 38.2% | 0.92 |
| Ours | 0.535 | **0.995** | **9.5998** | **34.7%** | 1.02 | **1.394** | **2.105** | **25.283** | **24.1%** | **0.97** |

## 4.3 ABLATION STUDY

As shown in Table 3, we compare our method with five main ablated versions: (1) **use VAE as encoder.** To demonstrate the stability of our method with different motion latent encodings, we replace the VQ-VAE with a VAE as the motion encoder. As shown in the table, using VAE as the motion encoder does not significantly affect the results. (2) **w/o motion encoder.** To highlight the importance of predicting motion latent codes rather than raw motions, we remove the VQ-VAE motion encoder and directly predicted the pose sequence. The results show that directly predicting the raw pose sequence significantly degrades the motion quality. (3) **use GPT.** We replace the diffusion model with GPT to predict the next token probabilities. The results show a decline in motion quality, with noticeably worse FID scores. (4) **w/o online decoder.** To validate the necessity of training a new online motion decoder, we directly apply the VQ-VAE decoder to decode the motion latent codes into motion sequences. Using the VQ-VAE decoder at each step results in discontinuous motion, which in turn leads to a worse FID score. (5) **w/o R in decoder.** We remove the root sequence **R** in Equation 5 as the input to the online motion decoder. Without **R**, the model can easily predict motions with the wrong root facing direction.

In summary, we demonstrate the importance of different components in our model. More ablation studies and visual results can be found in Appendix G and the supplementary materials.

## 5 APPLICATION: GENERATING REACTIVE MOTION WITH SPARSE SIGNALS

Introducing sparse control is essential for making our method practical in real-world applications, particularly in VR online interactive environments. In these settings, capturing detailed and dense input data can be challenging due to hardware limitations, computational costs, or user comfort. Sparse control addresses this by allowing the system to generate high-quality motion based on minimal input signals.

Table 4: **Quantitative results of generating reactive motion from sparse signals.** We compare our method with CAMDM. Among them, **bold** indicates the best results. ↓ means lower is better. → means closer to the real data is better. Our method outperforms the baseline in terms of all metrics.

| Methods | FID ↓ | | | Jitter → | FS → | Pos. Err. ↓ | Rot. Err. ↓ |
|---|---|---|---|---|---|---|---|
| | Per-frame | Per-transition | Per-clip | | | | |
| Real | - | - | - | 21.332 | 0.97 | - | - |
| CAMDM | 0.697 | 1.506 | 15.169 | 47.229 | 2.25 | 14.52 | 22.40 |
| Ours | **0.249** | **0.263** | **4.086** | **21.163** | **1.06** | **2.72** | **4.39** |

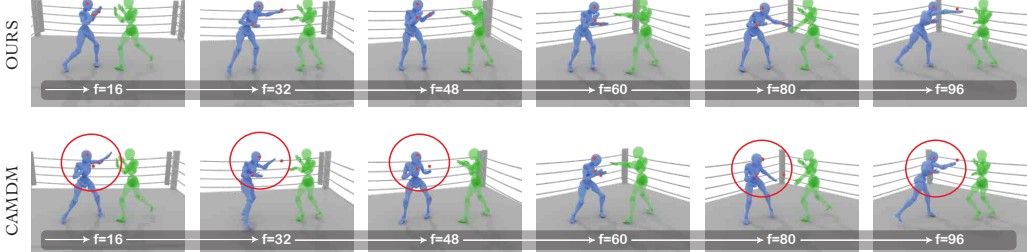

Figure 4: **Qualitative results of generating reactive motions from sparse signals.** We compare our method with CAMDM. Our approach successfully generates realistic motion while effectively adhering to the sparse signals (annotated by red dots in the figures). In contrast, CAMDM struggles to achieve the same level of responsiveness and accuracy, as shown in the red circles.

To demonstrate that our method is well-suited for VR online interactive environments, we also conducted experiments showing that it can be controlled by sparse signals. The sparse signals are the head and two-hand positions and rotations relative to the previous frame's agent root coordinate. To enable the controlling feature, we retrain the reaction policy by adding the sparse signals as conditions to the two transformer models in Figure 2. The loss and other training settings remain unchanged. We evaluate the quality of the generated motion using FID scores, motion jitter, foot sliding, and position and rotation errors to highlight the controllability of our approach. We compare our method with CAMDM (Chen et al., 2024), an auto-regressive method that generates diverse motions based on control signals. The results, presented in Table 4, show that our method consistently outperforms the baseline across all evaluated metrics. We also provide qualitative results in Figure 4 and in the supplementary materials.

## 6 DISCUSSION AND CONCLUSION

In this work, we have introduced a novel reaction policy that can be applied to two-character boxing interaction generation. The reaction policy includes a diffusion-based predictor for forecasting the next motion latent, paired with an online motion decoder optimized for the online generation. Our method effectively reduces error accumulation, enables real-time online generation, and can produce extended motion sequences. Additionally, it offers controllability through sparse signals, making it well-suited for online interactive environments.

However, several limitations to our approach should be addressed in future work. First, the current method is primarily designed for interactions between two individuals, and extending it to handle multi-person scenarios remains a challenge. Second, our method does not account for interactions with the environment or objects, which are essential for many real-world applications. Overcoming these limitations will be crucial for making the method more versatile and applicable to a wider range of scenarios.

## ACKNOWLEDGEMENTS

This work was partially supported by the NSFC (No. 62322207, No. 62172364, No. 62402427), Zhejiang Provincial Natural Science Foundation of China (No. LD25F030001), Ant Group Research Fund and Information Technology Center and State Key Lab of CAD&CG, Zhejiang University.

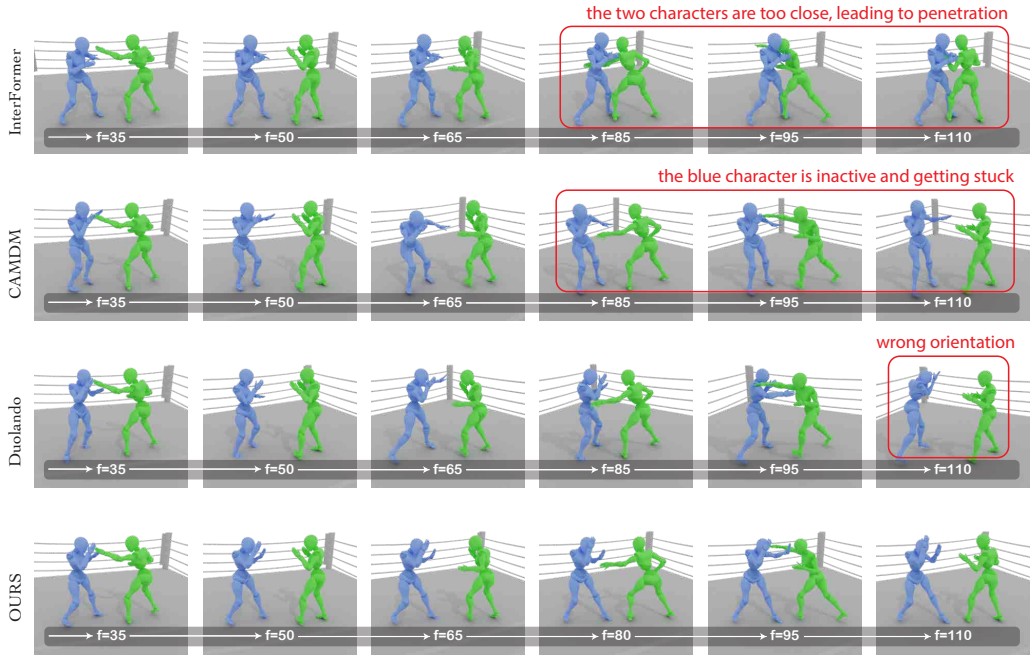

Figure 5: **Qualitative results of generating reactive motions.** Given the same ground truth opponent motion, InterFormer can produce reactive motion that is too close to the opponent, leading to penetration. CAMDM tends to get stuck, while Duolando may result in human motion with incorrect orientation after a certain period.

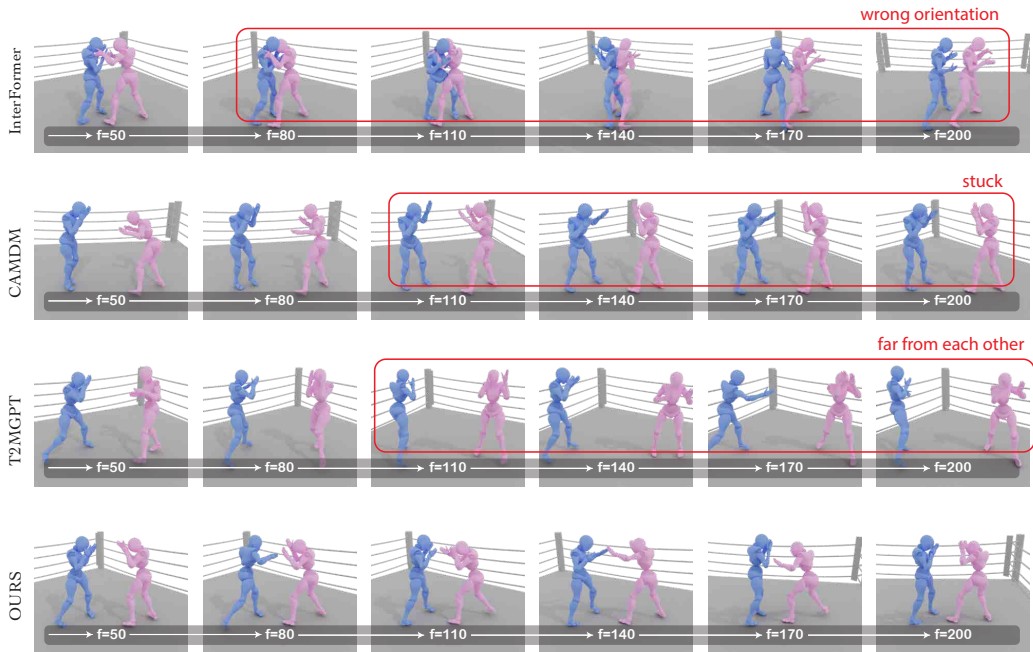

Figure 6: **Qualitative results of generating two-character motions.** Given the same initial four frames for both characters, InterFormer tends to produce human motion with incorrect orientation. CAMDM often results in the characters getting stuck, while T2MGPT can cause the two characters to drift apart due to accumulated errors.

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

# APPENDIX

## A DATASET DETAILS

### A.1 DATA COLLECTION DETAILS

Figure 7 illustrates our MoCap system. We use 12 high-resolution cameras with 120 FPS around the persons during our capture process. Figure 7 (a) and (b) shows the bird-view and the side-view of the camera positions. We apply 50 markers on one person as shown in Figure 7 (c) and (d). Figure 8 illustrates the motion capture setup and environment. The two actors wear motion capture suits with markers.

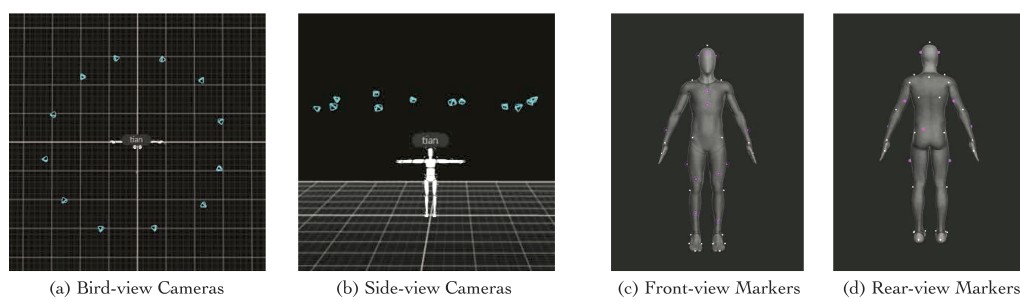

| (a) Bird-view Cameras | (b) Side-view Cameras | (c) Front-view Markers | (d) Rear-view Markers |

Figure 7: **Cameras and markers.** We have 12 cameras around the persons: (a) cameras from bird-view and (b) cameras from side-view. We put 50 markers on one person as in (c) front-view and (d) rear-view of markers.

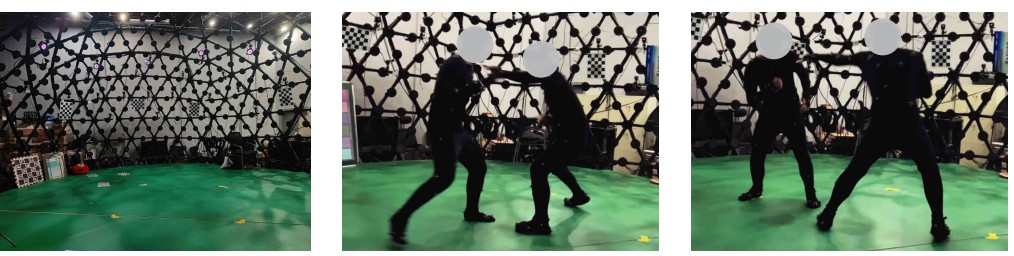

Figure 8: **Data collection.** During data collection, the actors wear motion capture suits with markers and perform boxing in the center of the area.

### A.2 DATASET STATISTICS

Table 5: **Dataset statics.** Types of movements in the DuoBox.

| # Sequences | Types of Movements |
| --- | --- |
| 25 | **Simple punches.** Jab, cross, (lead, rear) uppercut, (lead, rear) hook, defense, dodge, evade, counter, counterattack, parry. |
| 19 | **Combinations punches.** Jab-jab; jab-cross-rear uppercut; jab-lead uppercut; jab-cross-cross; jab-cross-lead uppercut; jab-cross-jab-rear uppercut; jab-cross-step-lead hook; jab-rear uppercut-lead upper cut-cross; jab-rear uppercut-jab-rear uppercut; jab face-cross body-jab face; jab-lead hook-cross; etc. |
| 72 | **Free sparring.** |

We recorded 116 sequences in total. The detailed statistics are presented in Table 5. In 25 sequences, the participants were instructed to perform simple punching actions. In 19 sequences, they were asked to execute specific boxing punch combinations [2]. In the remaining 72 sequences, they engaged in free-style boxing, simulating an actual match.

### A.3 ETHICAL CONSIDERATIONS

The participant group consisted of three male individuals aged 18-22, who were boxing enthusiasts recruited from a boxing club. All participants provided informed consent prior to participation and were fully informed of the research purpose. Since we used optical motion capture equipment, no photographic images of the participants were collected. To ensure privacy and confidentiality, all data was anonymized during processing, and any identifying information was removed.

## B ONLINE TWO-CHARACTER MOTION GENERATION ALGORITHM

We present a pseudo-algorithm in Algorithm 1 outlining the process of generating two-character motions using our reaction policy, as described in Section 3.3.

---

**Algorithm 1:** Online Two-Character Motion Generation with Reaction Policy

---

**Input:** Initial poses: $\{p_A^i\}_{i=1}^d$ for Agent, $\{p_O^i\}_{i=1}^d$ for Opponent; Reaction policy: $\mathcal{P}(\mathcal{O}, \mathcal{A})$;
      History buffer size: $W$; Frames per round: $d = 4$;
**Output:** Generated motions: $\{p_A^i\}_{i=d+1}^T$ for Agent, $\{p_O^i\}_{i=d+1}^T$ for Opponent;

Initialize history buffers: $\mathcal{B}_A \leftarrow \{p_A^i\}_{i=1}^d$, $\mathcal{B}_O \leftarrow \{p_O^i\}_{i=1}^d$;
Initialize time index: $t \leftarrow d$;
**for** *each round **until** $T$ frames are generated* **do**
    $\{p_A^{t+i}\}_{i=1}^d \leftarrow \mathcal{P}(\mathcal{B}_O, \mathcal{B}_A)$;
    $\{p_O^{t+i}\}_{i=1}^d \leftarrow \mathcal{P}(\mathcal{B}_A, \mathcal{B}_O)$;
    Append $\mathcal{B}_A$ with $\{p_A^{t+i}\}_{i=1}^d$;
    Append $\mathcal{B}_O$ with $\{p_O^{t+i}\}_{i=1}^d$;
    **if** $|\mathcal{B}_A| > W$ **then**
        | Remove the oldest element from $\mathcal{B}_A$ and keep length of $W$;
    **end**
    **if** $|\mathcal{B}_O| > W$ **then**
        | Remove the oldest element from $\mathcal{B}_O$ and keep length of $W$;
    **end**
    Update time index $t \leftarrow t + d$;
**end**

---

## C DETAILS OF NETWORK ARCHITECTURE

The VQ-VAE network architecture follows Zhang et al. (2023). Both the encoder and decoder consist of 1D convolution and residual blocks. Temporal downsampling is achieved using convolutions with stride 2, while upsampling is performed via nearest interpolation. The downsampling and upsampling are applied twice, resulting in a total downsampling rate of $d = 2^2 = 4$.

The two transformer models in Figure 2 use 8 transformer layers with 4 heads. The diffusion model consists of a single-layer MLP. We also tried various types and numbers of layers of the MLP in the diffusion process, but we found that single-layer MLP is sufficient and effective to produce lower FIDs. An additional ablation towards the diffusion network design can be found at Appendix G.3. During training, the next latent predictor and the online motion decoder are trained together. The output next latent shaped with (B, L, C) are reshaped to (B×L, C) and passed to the online motion decoder.

---

[2] https://www.youtube.com/watch?v=V59fcX1YC7E

## D  IMPLEMENTATION OF THE BASELINES

Here, we provide the details of the baseline implementations used for comparison with our method, which is evaluated against several state-of-the-art motion generation baselines. (1) **InterFormer.** InterFormer is an online deterministic reactive motion prediction method. We retrain this model on our dataset. (2) **CVAE-AR.** We designed a CVAE-based method specifically for comparison, utilizing the standard CVAE pipeline with KL and reconstruction loss. The encoder and decoder architectures follow those from TEMOS (Petrovich et al., 2022). (3) **CAMDM.** CAMDM is an auto-regressive diffusion model that generates motion sequences with root control. We adapt the model by using the opponent's motion in place of root control and retrain it on our dataset. (4) **T2MGPT-online.** T2MGPT is originally a text-to-motion model. To make it an online method, we modify it to immediately decode each newly generated token into raw motion using a VQ-VAE decoder. The opponent's motion was used as input instead of text. (5) **Duolando-offline.** Duolando is designed for the duet dance generation. We adapt its network architecture but exclude its RL finetuning for the comparison.

## E  INFERENCE SPEED ANALYSIS

We also test the inference speed and the results are shown in Table 6. Although our model is the slowest, it can achieve real-time inference and the lowest FID scores.

Table 6: **Inference speed.** We compare the inference time per frame with other methods. All methods are tested on a single Nvidia RTX 4090 GPU.

| Methods | InterFormer | CVAE-AR | CAMDM | T2MGPT-online | Duolando-offline | Ours |
|---|---|---|---|---|---|---|
| Inference time per frame (ms) ↓ | 12 | **2** | 5 | 17 | 5 | 22 |
| Reactive per-clip FID ↓ | 15.061 | 26.010 | 28.503 | 30.159 | 13.606 | **9.599** |
| Two-character per-clip FID ↓ | 47.194 | 92.978 | 70.416 | 122.463 | - | **25.283** |

## F  MOTION DIVERSITY ANALYSIS

For each sequence input in the test set, we generate 10 samples and calculate Average Pairwise Distance (APD) scores to evaluate the motion diversity Yuan & Kitani (2020). The results are shown in Table 12. A greater value means the generated motions are more diverse. Among the methods, InterFormer is deterministic and therefore lacks diversity. We observed that other baselines show greater diversity, primarily due to their higher levels of error accumulation.

Table 7: **Motion diversity.** We compare the generated motion diversity with baselines.

| Methods | InterFormer | CVAE-AR | CAMDM | T2MGPT-online | Duolando-offline | Ours |
|---|---|---|---|---|---|---|
| Reactive APD | 0.000 | 2.375 | 3.385 | 3.728 | 1.726 | 2.154 |
| Reactive per-clip FID ↓ | 15.061 | 26.010 | 28.503 | 30.159 | 13.606 | **9.599** |
| Two-character APD | 0.000 | 4.431 | 6.300 | 7.143 | - | 3.709 |
| Two-character per-clip FID ↓ | 47.194 | 92.978 | 70.416 | 122.463 | - | **25.283** |

## G  MORE ABLATION STUDY

### G.1  ABLATION STUDY OF THE TRAINING PROCESS

We also conduct a more specific ablation study on the training process. We compare our method with two variants: (1) **split training**: we split the training of the next latent predictor and the online motion decoder. The predicted next latent is not fed into the online motion decoder while training. (2) **stop gradient**: we stop the gradient flow from the online motion decoder to the next latent

predictor (the diffusion model). The results are shown in Table 8. From the table, we observe that split training outperforms our method in reactive per-frame metrics, RO, and FS, but performs poorly in the two-character scenarios. We can conclude that supervising the online motion decoder with the ground truth pose allows for joint optimization of both the next latent predictor and the online motion decoder, leading to better overall performance compared to the two ablation methods.

Table 8: **Ablation study of the training process.** We conduct an ablation study towards the training process. Among them, **bold** indicates the best results. ↓ means lower is better. → means closer to the real data is better.

| Methods | Reactive | | | | | Two-character | | | | |
|---|---|---|---|---|---|---|---|---|---|---|
| | FID↓ | | | RO→ | FS→ | FID↓ | | | RO→ | FS→ |
| | Per-frame | Per-trans. | Per-clip | | | Per-frame | Per-trans. | Per-clip | | |
| Real | - | - | - | 24.7% | 0.97 | - | - | - | 24.7% | 0.97 |
| split training | **0.522** | 1.203 | 10.004 | **33.4%** | **0.96** | 1.618 | 2.679 | 29.574 | 19.8% | 0.94 |
| stop gradient | 0.536 | 1.190 | 11.064 | 36.3% | 1.01 | 1.822 | 3.589 | 34.797 | 26.1% | 0.98 |
| Ours | 0.535 | **0.995** | **9.5998** | 34.7% | 1.02 | **1.394** | **2.105** | **25.283** | **24.1%** | **0.97** |

## G.2 Ablation Study of the Visible Window Size

We conduct an ablation study on the visible past window size $W$ stated in Section 3.1. We compare our setting $W = 60$ with $W = 40$ and $W = 80$. The results are shown in Table 9. Finally, we select $W = 60$ for its better performance on FID scores.

Table 9: **Ablation study of the visible window size.** We conduct an ablation study towards the visible window size W. Among them, **bold** indicates the best results. ↓ means lower is better. → means closer to the real data is better.

| Methods | Reactive | | | | | Two-character | | | | |
|---|---|---|---|---|---|---|---|---|---|---|
| | FID↓ | | | RO→ | FS→ | FID↓ | | | RO→ | FS→ |
| | Per-frame | Per-trans. | Per-clip | | | Per-frame | Per-trans. | Per-clip | | |
| Real | - | - | - | 24.7% | 0.97 | - | - | - | 24.7% | 0.97 |
| $W = 40$ | 0.610 | 1.416 | 11.631 | **33.7%** | 0.99 | 1.982 | 3.483 | 34.530 | 19.0% | 0.94 |
| $W = 80$ | 0.583 | 1.038 | 10.994 | 35.1% | 1.02 | 1.625 | 2.680 | 30.319 | 21.2% | **0.97** |
| $W = 60$ (Ours) | **0.535** | **0.995** | **9.5998** | 34.7% | 1.02 | **1.394** | **2.105** | **25.283** | **24.1%** | **0.97** |

## G.3 Ablation Study of the Diffusion Network Design

As stated in Appendix C, we explored various network architectures during the design phase. However, we ultimately found that a single-layer MLP not only offers faster inference but also achieves a lower FID. To support our statement, we conduct an ablation study on the diffusion network design. We compare our single-layer MLP with five variants: (1) 2-MLP: using 2 layers of MLP. (2) 4-MLP: using 4 layers of MLP. (3) 1-ResNet: using 1 layer of ResNet. (4) 2-ResNet: using 2 layers of ResNet. (5) 4-ResNet: using 4 layers of ResNet. The ResNet is adopted from Stable Diffsuion[3], and we make some modifications: (1) we use linear norm, (2) we use linear layer instead of convolution. The results are shown in Table 10.

From the table, we can conclude that the 1-MLP design can achieve better performance generally. We considered several possible reasons to explain this phenomenon. First, more complex networks may overfit the training dataset faster, potentially leading to poorer performance on the test dataset. Second, since the diffusion denoising process involves 1000 steps, simpler transformations at each step might already suffice to achieve the desired results.

---

[3] https://github.com/Stability-AI/generative-models

Table 10: **Ablation study of the diffusion network design.** N-MLP means using N layers of MLPs. N-ResNet means using N layers of ResNet. Our method use the 1-MLP design.

| Methods | Reactive | | | | | Two-character | | | | |
|---|---|---|---|---|---|---|---|---|---|---|
| | FID↓ | | | RO→ | FS→ | FID↓ | | | RO→ | FS→ |
| | Per-frame | Per-trans. | Per-clip | | | Per-frame | Per-trans. | Per-clip | | |
| Real | - | - | - | 24.7% | 0.97 | - | - | - | 24.7% | 0.97 |
| 2-MLP | 0.812 | 1.682 | 14.525 | 35.2% | 0.90 | 2.111 | 3.594 | 35.970 | 20.5% | 0.87 |
| 4-MLP | 0.763 | 1.472 | 14.258 | 35.3% | 0.92 | 2.030 | 3.474 | 36.831 | 21.2% | 0.86 |
| 1-ResNet | 0.656 | 1.324 | 12.653 | 34.5% | 0.94 | 1.727 | 3.000 | 32.142 | 21.5% | 0.92 |
| 2-ResNet | 0.624 | 1.235 | 11.650 | 33.4% | **0.95** | 1.750 | 2.750 | 31.887 | 20.9% | 0.93 |
| 4-ResNet | 0.703 | 1.518 | 13.610 | **31.4%** | 0.93 | 1.925 | 3.256 | 34.499 | 18.5% | 0.89 |
| 1-MLP (Ours) | **0.535** | **0.995** | **9.5998** | 34.7% | 1.02 | **1.394** | **2.105** | **25.283** | **24.1%** | **0.97** |

# H APPLYING OUR METHOD TO THE INTER-X DATASET

We selected three actions from the Inter-X with varying contact frequencies—chat, kick, and dance—in increasing order of contact frequency. We use the first 22 joints from the SMPLX skeleton. The network and the training protocols remain unchanged. For the two-character FID score, we calculate the per-frame, per-transition, and per-clip features for each individual, alternating between the "agent" and the "opponent". We removed the RO metric, because in actions unlike boxing, the two characters do not necessarily need to always face each other.

Results are shown in Table 11. From the tables, we can observe that our method generally outperforms the selected baselines, demonstrating its ability to generalize to different types of motions effectively.

Table 11: **Comparison with baselines on the Inter-X dataset.** We compare our method with CAMDM and T2MGPT on the Inter-X dataset.

(a) Quantitative results on the Inter-X dataset with action category of "Chat".

| Methods | Reactive | | | | Two-character | | | |
|---|---|---|---|---|---|---|---|---|
| | FID↓ | | | FS→ | FID↓ | | | FS→ |
| | Per-frame | Per-trans. | Per-clip | | Per-frame | Per-trans. | Per-clip | |
| Real | - | - | - | 0.20 | - | - | - | 0.20 |
| CAMDM | 1.693 | 0.405 | 21.024 | 0.67 | 1.472 | 0.357 | 18.931 | 0.70 |
| T2MGPT-online | 1.241 | 0.165 | 17.757 | 1.32 | 2.001 | 0.207 | 28.007 | 1.60 |
| Ours | **1.163** | **0.120** | **15.372** | 0.30 | **1.148** | **0.10** | **15.215** | 0.31 |

(b) Quantitative results on the Inter-X dataset with action category of "Kick".

| Methods | Reactive | | | | Two-character | | | |
|---|---|---|---|---|---|---|---|---|
| | FID↓ | | | FS→ | FID↓ | | | FS→ |
| | Per-frame | Per-trans. | Per-clip | | Per-frame | Per-trans. | Per-clip | |
| Real | - | - | - | 0.76 | - | - | - | 0.76 |
| CAMDM | 1.127 | **0.520** | 18.427 | 1.96 | 1.209 | 0.604 | 19.484 | 1.99 |
| T2MGPT-online | 1.465 | 0.883 | 22.283 | 1.89 | 1.770 | 0.914 | 26.250 | 1.93 |
| Ours | **0.757** | 0.593 | **12.861** | 0.79 | **0.750** | **0.558** | **12.828** | 0.83 |

(c) Quantitative results on the Inter-X dataset with action category of "Dance".

| Methods | Reactive | | | | Two-character | | | |
|---|---|---|---|---|---|---|---|---|
| | FID↓ | | | FS→ | FID↓ | | | FS→ |
| | Per-frame | Per-trans. | Per-clip | | Per-frame | Per-trans. | Per-clip | |
| Real | - | - | - | 0.68 | - | - | - | 0.68 |
| CAMDM | 2.239 | 1.251 | 34.199 | 1.02 | 2.128 | 1.192 | 34.548 | 1.21 |
| T2MGPT-online | 1.185 | **0.404** | 22.380 | 1.59 | 2.714 | **0.275** | 40.943 | 1.77 |
| Ours | **1.169** | 0.653 | **21.300** | 0.58 | **1.065** | 0.435 | **18.646** | 0.53 |

## I   PENETRATION ANALYSIS

Our skeleton representation is based on positions and rotations exported from motion capture software. In the main paper, we visualize the predicted rotations by mapping them onto an "Xbot" model for demonstration purposes. Note that the Xbot skeleton differs from the one we use in our method, which makes penetration evaluations based on Xbot less reliable. As a result, penetration cannot be directly evaluated at the mesh level.

To evaluate penetration between skeletons, we approximate the bones using triangular prisms with a distance from the centroid to a vertex $= 5\ cm$ (as shown in Figure 9) and calculate penetration frame by frame between the two body meshes by using $trimesh.Trimesh.intersection$ function. Since boxing typically involves instantaneous contact, the proportion of frames with penetration is expected to be very low. Therefore, we report the total number of frames with penetration across the entire test set, and the mean penetration volume in $cm^3$.

From the table, we can see that in the reactive setting, our method achieves comparable results to Duolando-offline, which predicts root positions relative to the opponent's root. In the two-character setting, our method achieves a lower mean penetration volume compared to all the baselines, demonstrating its effectiveness in avoiding penetration.

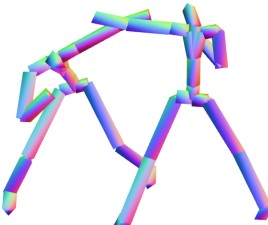

Figure 9: **Skeleton mesh that is used to compute penetration.** We create approximated skeleton meshed using triangular prism with distance from the centroid to a vertex $= 5\ cm$.

Table 12: **Penetration analysis.** We measure the number of penetration frames and mean penetration volume (in $cm^3$) in the test set. ↓ means lower is better.

| Settings | Metrics | InterFormer | CVAE-AR | CAMDM | T2MGPT-online | Duolando-offline | Ours | GT |
|---|---|---|---|---|---|---|---|---|
| Reactive | # Penetration Frames | 1969 | 1571 | 860 | 1846 | 417 | 1084 | 141 |
| | Mean Penetration Volume ↓ | 200.43 | 237.51 | 200.16 | 218.78 | **118.51** | 161.35 | 21.37 |
| Two-character | # Penetration Frames | 2617 | 1858 | 469 | 962 | - | 757 | 141 |
| | Mean Penetration Volume ↓ | 222.83 | 218.15 | 129.14 | 365.21 | - | **111.84** | 21.37 |

## J   CONTACT IMPACT ANALYSIS

Table 13: **Cotact impact analysis.** All contact analysis is conducted under the condition that one's hand is close enough to the other. $\#OF$ represents the number of frames that the opponent moves forward. $\#OF\_AB$ represents the number of frames that the agent moves backward when the opponent moves forward. $Ratio = \frac{\#OF - AB}{\#OF}$. → means closer to the real data is better.

| Settings | Metrics | InterFormer | CVAE-AR | CAMDM | T2MGPT-online | Duolando-offline | Ours | GT |
|---|---|---|---|---|---|---|---|---|
| Reactive | # OF | 1163 | 997 | 532 | 1145 | 402 | 761 | 243 |
| | # OF-AB | 615 | 505 | 260 | 613 | 272 | 397 | 125 |
| | Ratio → | 52.80% | 50.60% | 48.80% | 53.50% | 23.30% | **52.10%** | 51.40% |
| Two-character | # OF | 1179 | 1484 | 371 | 539 | - | 816 | 243 |
| | # OF-AB | 921 | 716 | 193 | 267 | - | 420 | 125 |
| | Ratio → | 78.10% | 48.20% | 52.00% | 49.50% | - | **51.40%** | 51.40% |

To analyze the impact of sparse contact on motion generation, we first detect contact and then assess whether the agent moves backward or forward in response. Specifically, we identify contact by calculating the distance between one character's skeleton mesh and the other's hand joints. If the distance is less than $5\ cm$, we consider that the hand has successfully made contact with the other. This could include the opponent's hand hitting any part of the agent or vice versa. For these identified contact frames, we examine whether the agent's root moves backward when the opponent's root moves forward and compute the proportion of such occurrences. For each frame, we switch the roles of "agent" and "opponent". Results are shown in Table 13. A higher alignment with the ground truth indicates better performance.

From the table, we observe that our method achieves performance closer to the ground truth, demonstrating its ability to produce more realistic reactions.

