# OpenReview forum: "Ready-to-React: Online Reaction Policy for Two-Character Interaction Generation"
_ICLR.cc/2025/Conference — ICLR 2025 Poster_

### Official Review · Reviewer_TLUt · 2024-10-21

**Soundness:** 3
**Presentation:** 3
**Contribution:** 3
**Rating:** 6
**Confidence:** 3

**Summary:**

This paper introduces a novel approach to generating real-time interactions between two characters. The proposed "Ready-to-React" method focuses on creating dynamic, continuous motions for each character based on their counterpart’s actions. The key innovation lies in incorporating a diffusion head into an auto-regressive model, which helps mitigate error accumulation and improves the naturalness of long-term interactions. The approach is demonstrated on a boxing task, where it outperforms existing methods in generating coherent and plausible interactions over extended sequences. This technique has potential applications in VR and online interactive environments.

**Strengths:**

1. Experimental adequacy: The experimental result is persuasive compared with existing baselines and ablation studies are sufficient to prove the effectiveness of the proposed method.
2. The insight into error accumulation mitigation in two-character motion generation is interesting.

**Weaknesses:**

1. Lack of significance: The background of this work is too narrow and I do not think it is important to study. Also, from my perspective, this method is for interaction generation. However, there has been much work studying video generation which seems to be a lot more complicated than this paper.
2. Lack of novelty: The proposed method simply uses diffusion to generate representations instead of directly generating actions which is lack of novelty and contribution.
3. This method is only tested in one dataset which is not sufficient although you mention this scene is a difficult one.

**Questions:**

1. You mention in the introduction that "incorporating a diffusion head into an auto-regressive model, which can respond to the counterpart’s motions in a streaming manner while ensuring the naturalness and diversity of the motions". How can this method help diversity?
2. Can you use a few baselines that are not deep learning, such as those executed by programmed programs? Because in this scene (such as ), I think pre-programmed is quite smooth.

---

> ### Author Response · Authors · 2024-11-22
>
> **Narrow background.**
>
> We strongly disagree with the statement that "the background of this work is too narrow and I do not think it is important to study." As noted by **R2-T1pE**, the task holds significant importance in two-person interaction motion generation. Additionally, **R1-odxr** and **R3-qtLW** emphasized the value and contribution of our proposed dataset to the community. Furthermore, we demonstrate that our method can be effectively generalized to other datasets (e.g., Inter-X) in the general response and $\textcolor{green}{\text{Appendix H}}$.
>
> **Lack of novelty.**
>
> Our primary focus is on two-character animation generation, and the core contribution lies at a conceptual level in addressing a task that better aligns with human interactive processes. This provides a fresh perspective and technical insight into motion generation for interactive scenarios.
>
> As the reviewer **R3-qtLW** noted, the method is "simple yet effective and versatile", which aligns with our goal of real-time generation. We deliberately opted for a simpler architecture to achieve this objective.
>
> Furthermore, this work explores the use of diffusion in real-time interaction tasks, a direction that has not been explored before, offering new insights into the field. The results demonstrate significant improvements over baseline methods, highlighting the effectiveness of our approach and its potential to inspire future research.
>
> **Generalize to other datasets.**
>
> Please refer to the general response: additional results on the Inter-X dataset and $\textcolor{green}{\text{Appendix H}}$.
>
> **Clarify how the diffusion head helps motion diversity.**
>
> The term "ensure" in the introduction (L72) is intended to convey "preserve", rather than "help". Previous works (e.g., MDM [1]) have demonstrated that conditional diffusion can enhance diversity, providing evidence for this aspect.
>
> **Pre-programmed baseline.**
>
> We speculate that you are discussing motion matching methods commonly used in games. Motion matching ([motion matching example](https://github.com/orangeduck/Motion-Matching)) relies on a set of manually designed features to match motions from a predefined library. It is primarily used to generate motions based on existing user inputs. In contrast, our method is capable of directly generating interactive boxing motions for two characters without requiring any user input. In games, automated boxing agents often rely on extensive rules to generate commands, which are then used to drive motion matching and produce specific character motions. We have not found any open-sourced baselines specifically designed for this task. If you could recommend one, we would be happy to compare our method with it.
>
> ---
> [1] Tevet, Guy, et al. "MDM: Human Motion Diffusion Model". In ICLR. 2023.

---

> > ### Comment · Reviewer_TLUt · 2024-11-22
> >
> > Thank you for the response and sorry for the misunderstanding. I've looked at the other reviews/rebuttal responses; I think the explanation and improved clarity make the paper a motivated and solid one to be accepted.
> >
> > I'm not familiar enough with this area so I will raise my score to 6, but I think this paper should be accepted.

---

> > > ### Author Response · Authors · 2024-11-23
> > >
> > > Thank you for taking the time to review our work and for reading the other reviews and rebuttal responses. We sincerely appreciate your recommendation for acceptance.

---

### Official Review · Reviewer_qtLW · 2024-10-24

**Soundness:** 3
**Presentation:** 3
**Contribution:** 3
**Rating:** 8
**Confidence:** 4

**Summary:**

The paper present a new method for human reactive motion generation. The method contains a history encoder to encode historic motion using VQ-VAE or MLP, a latent predictor using Transformer encoder and diffusion and a latent decoder using Transformer decoder. The proposed architecture can be use for reactive motion generation but also for two persons interaction generation and support long term generation (1800 frames). The proposed method outperforms the state of the art quantitatively and qualitatively on a new boxing dataset containing more than 1 hour of 3D motions.

**Strengths:**

- The method is relatively simple yet performs very well and is very versatile as it can be used for reactive motion generation but also for two persons interaction generation.

- The method is efficient even for long term generation.

- The paper presents extensive experiments and ablation.

- The authors propose a new interaction (boxing) dataset with long sequence. This is very welcome due the the scarcity of long interaction datasets.

- The authors provided qualitative results in video format.

**Weaknesses:**

- The experiments are performed on only one dataset and this dataset is the new one proposed in the paper. It would have been better to also have experiment on another dataset e.g one of those used for Duolando or Interformer.

- The paper does not precise if the dataset will be released to the public.

- While I understood the reasoning behind using diffusion with MLP for the latent predictor, I am surprised that it works so well with only a single layer. What was the intuition behind the choice of this single layer MLP, did the authors try other configurations for the denoising network ? This would have been interesting to explore this more in details.

- A few things are still not clear to me in the methodology:

    - Line 176: what is the reasoning behind the exclusion of the y-axis when extracting r_off and r_dir?

    - Why does the opponent input doesn't contain r_off and r_dir when the agent input uses them?

    - Also on the subject of differences between agent and opponent, why use a VQ-VAE for the AGENT but a MLP for the opponent?

    - How does the two-character motion generation works exactly? Does this work in a auto-regressive manner? i.e. generate agent motion T -> generate opponent motion T+1 -> generate agent motion T+1->... . Does it need two separately trained networks (one for agent and one for opponent) or only one ? More details are necessary.

- For the two-character motion generation experiments the authors do not compare with Duolando. This is understandable since it was build to generate the motion of only one person in an interaction. However this seems to also be the case for Interformer yet the authors were able to compare against it. Were some modifications done to the network architecture or to the training/inference protocol to make it possible? this should be clarified.

**Questions:**

- In figure 2 "Tranformer" is used in both the latent predictor and the motion decoder. from what I understood of the text there is only a Transformer encoder in the latent predictor and a transformer decoder in the motion predictor. If it is the case the authors should update the figure to reflect this. Using the term "Transformer" alone make it seems it contains both encoder and decoder.

- Notation error: "d" is used both for the down-sampling factor (line 198) and for the initial number of frame (line 248)

---

> ### Author Response · Authors · 2024-11-22
>
> **Generalize to other datasets.**
>
> Please refer to the general response: additional results on the Inter-X dataset and $\textcolor{green}{\text{Appendix H}}$.
>
> **The dataset will be released** as stated at L48 in the main paper.
>
> **The design choice of single-layer MLP in diffusion.**
>
> As stated in Appendix C (L795–L797), we explored various network architectures during the design phase. However, we ultimately found that a single-layer MLP not only offers faster inference but also achieves a lower FID. Recently, we surprisingly found a [blog](https://theorangeduck.com/) that has a similar conclusion to us. They found that an auto-encoder with 2 layers performs better than 8 layers.
>
> We also add an ablation study towards the diffusion network layers. Below are the explanations of the ablated versions and results in the two-character setting. We also updated the whole table and details in $\textcolor{green}{\text{Appendix G.3}}$.
>
> - **1-MLP(Ours)**: our single layer MLP.
> - **N-MLP**: using N layers of MLP.
> - **N-ResNet**: using N layer of ResNet. The ResNet is adopted from [Stable Diffusion](https://github.com/Stability-AI/generative-models/blob/1659a1c09b0953ad9cc0d480f42e4526c5575b37/sgm/modules/diffusionmodules/openaimodel.py#L242). We make some modifications: (1) we use linear norm, (2) we use linear layer instead of convolution.
>
> |Variants|Per-frame FID &#8595;|Per-trans FID &#8595;|Per-clip FID &#8595;|RO &#8594;|FS &#8594;|
> |---|---:|---:|---:|---:|---:|
> | GT|-|-|-|24.70 %|0.97|
> |1-MLP, Ours|**1.394**|**2.105**|**25.283**|**24.10 %**|**0.97**|
> |2-MLP|2.111|3.594|35.970|20.50 %|0.87|
> |4-MLP|2.030|3.474|36.831|21.20 %|0.86|
> |1-ResNet|1.727|3.000|32.142|21.50 %|0.92|
> |2-ResNet|1.750|2.750|31.887|20.90 %|0.93|
> |4-ResNet|1.925|3.256|34.499|18.50 %|0.89|
>
> From the table, we can conclude that the 1-MLP design can achieve better performance. We considered several possible reasons to explain this phenomenon. First, more complex networks may overfit the training dataset faster, potentially leading to poorer performance on the test dataset. Second, since the diffusion denoising process involves 1000 steps, simpler transformations at each step might already suffice to achieve the desired results.
>
> **Clarify some details in the method.**
> - **Reason of excluding the y-axis of r_off and r_dir in agent motion representation.**
> Since we are dealing with an auto-regressive problem, we place significant emphasis on addressing error accumulation. If we define the root as the position and rotation of the pelvis joint, it may result in vertical movement along the y-axis (gravity direction). However, boxing takes place on flat ground without elevation changes. Therefore, defining the root as the projection of the pelvis onto the ground helps mitigate this issue. Additionally, we align the x and z axes of the root coordinate system parallel to the ground, transforming root motion into purely 2D planar movement. This ensures that the pitch orientation of the root does not accumulate errors over time.
>
> - **Reason of excluding the opponent's r_off and r_dir as input** is that r_off is determined by the opponent’s pelvis position, and r_dir is determined by the opponent's shoulder and hip joints. Therefore, including this information does not significantly impact the agent’s ability to understand the opponent’s movements. We also experimented with adding the opponent’s r_off and r_dir as conditions, and it showed no noticeable effect.
>
> - **Reason of encoding the agent by VQ-VAE while encoding the opponent by MLP.**
> There are two main reasons. The first is to preserve the relative positional information of the opponent. Encoding the opponent using the same VQ-VAE trained for a single character would result in the loss of global information relative to the agent, which is crucial for accurate interaction modeling.
> To incorporate this information effectively, we were inspired by recent VLM models (e.g., LLaVA), which use fully connected layers to inject additional conditions into transformers. This approach ensures the opponent’s information is integrated efficiently into the model.
>
> - **The process of the two-character motion generation** is conducted in an auto-regressive manner. A pseudo-algorithm is provided in Appendix B. This process does not require two separate trained networks; both agents share the same network.
>
> **Training and inference protocol of Interformer.**
>
> Interformer generates motions auto-regressively by predicting the next raw pose of the agent at each step. As a result, its training process does not require modification. We simply applied our inference protocol during the inference stage.
>
> **The transformer notations in Figure 2** have been updated to avoid ambiguity in the $\textcolor{green}{\text{revised main paper}}$.
>
> **The notation of "d"** has been clarified to distinguish between the down-sampling factor and the initial number of frames in the $\textcolor{green}{\text{revised main paper}}$.

---

> > ### Comment · Reviewer_qtLW · 2024-11-25
> > **review after rebuttal**
> >
> > I thanks the authors for their extensive rebuttal. After reviewing it, I find that the authors answered all of my concerns. I see no reason to change my ratting.

---

> > > ### Author Response · Authors · 2024-11-25
> > >
> > > Thank you for taking the time to review our work and for your thoughtful feedback. We are glad to hear that our rebuttal addressed all of your concerns.

---

### Official Review · Reviewer_T1pE · 2024-10-28

**Soundness:** 3
**Presentation:** 3
**Contribution:** 3
**Rating:** 8
**Confidence:** 4

**Summary:**

This work proposes an autoregressive diffusion model to generate motion for human boxing matches. The main innovation lies in an autoregressive policy that handles both the human actor's and opponent's policy concurrently, ensuring there is an "action" and "reaction" aspect for each player. To train the proposed method, a new dataset, DuoBox, is collected. Results show that the proposed method can generate long and realistic boxing motions.

**Strengths:**

- This work addresses an important problem in two-person interaction motion generation: handling interaction. The proposed autoregressive approach is sound and intuitive.
- The proposed online motion decoder is a well-thought-out solution for applying VQ-VAE styled motion representation in the autoregressive motion generation scheme.
- The special focus on root orientation prediction is crucial for the boxing task, based on the ablation.
- The generated motion is of high quality and visually pleasing. Quantitative results also show that the generated motion is of high quality and outperforms prior arts.
- The study on sparse VR input is a welcome addition, showcasing the method's applicability in animation.

**Weaknesses:**

- At video 0:45 (of the gray box), it is clear that the reaction of the other character can contain large penetration and unrealistic interaction. These should be properly handled. In fact, from watching the video, I would say that while the motion quality of the two characters is high, the "reaction" part is not well-demonstrated. There are many cases where punches are thrown, but the opponent does not evade and has no discernible reaction.
- I feel that given the study's scope, the title should reflect the focus on "boxing". Many proposed techniques (such as root orientation) focus on boxing. Extending to two-person sports (fencing, dancing, etc.) could be an interesting direction, but the proposed method has not been shown to handle generic two-person interactions.

**Questions:**

Does the model generalize beyond boxing? There have been several two-person interaction datasets proposed (e.g., [1]).
[1] Xu, Liang, et al. "Inter-x: Towards versatile human-human interaction analysis." *Proceedings of the IEEE/CVF Conference on Computer Vision and Pattern Recognition*. 2024.

---

> ### Author Response · Authors · 2024-11-22
>
> **Generalize to other datasets.**
>
> We follow your suggestion to test our method on the Inter-X dataset. Please refer to the general response: additional results on the Inter-X dataset and $\textcolor{green}{\text{Appendix H}}$.
>
> **Include "boxing" in the title.**
>
> We acknowledge that the experiments in the main paper were conducted solely on boxing. However, our method was not specifically designed for boxing. Regarding the design based on root orientation, we believe it is a general consideration applicable to all two-person tasks.
>
> We have reported the performance of our method on the Inter-X dataset. If you and other reviewers still feel the title should explicitly reflect this focus, we are open to revising it accordingly.
>
> **Suboptimal reaction movements.**
>
> We acknowledge that there is still room for improvement in the reactive movements. To further analyze the impact of contact on the generated results,  we report the agent’s movement relative to the opponent’s movement following reviewer **R1-odxr**’s suggestions. Details and results are shown in $\textcolor{green}{\text{Appendix J}}$.

---

> > ### Comment · Reviewer_T1pE · 2024-12-02
> > **Official Response by Reviewer**
> >
> > Thank you for the response! I appreciate the added information (Appendix J) and the results on Inter-X. I do feel like the title should somehow reflect boxing or competitive two-person sports. Qualitative results on the Inter-X dataset would also be appreciated.
> >
> > I recommend accepting this paper and will keep my score.

---

> > > ### Author Response · Authors · 2024-12-02
> > >
> > > Thank you for your thoughtful feedback and valuable suggestions regarding the title and adding Inter-X visual results. Unfortunately, as the paper revision deadline has passed, we are unable to make further changes to the PDF at this stage. We greatly appreciate your time in reviewing our work and are glad that our rebuttal addressed all your concerns. Thank you for recommending our paper for acceptance!

---

### Official Review · Reviewer_odxr · 2024-11-04

**Soundness:** 3
**Presentation:** 3
**Contribution:** 2
**Rating:** 6
**Confidence:** 5

**Summary:**

This paper introduces a new two-person boxing dataset to support three tasks: (1) reactive motion generation, (2) two-character interaction generation, and (3) long-term two-character interaction generation. The authors demonstrate that their method can generate motions lasting over one minute and outperforms existing methods across various tasks. Interestingly, the authors verify that their approach can be used in VR scenarios.

**Strengths:**

1. The proposed DuoBox dataset is a strong dataset for two-person interaction, which is highly beneficial to the community. It provides a new test benchmark for three novel tasks.

2. The method proposed in the paper can generate impressive results, significantly outperforming GPT-based methods and capable of generating longer action sequences.

**Weaknesses:**

1. Given that there is not much actual contact between individuals in this scenario, with contact occurring only at specific moments, have the authors analyzed how the occurrence of such contact influences subsequent motion generation?

2. Since the visualizations provided by the authors seem to be based on some kind of simulation environment, have the authors quantified metrics such as penetration (clipping) or foot sliding to assess the physical plausibility of the generated results?

**Questions:**

This paper investigates a very interesting problem. However, the paper does not consider the modeling of contact information during two-person interactions, which seems to limit its application in real robots. Nevertheless, the authors demonstrate the potential of their approach in VR applications. Overall, I have given my current score based on these considerations.

**Details Of Ethics Concerns:**

The paper involves data collection related to human subjects, and the authors need to provide additional details regarding this aspect. This should include information on how the data was collected, the demographic characteristics of the participants, any ethical considerations, consent procedures, and measures taken to ensure the privacy and confidentiality of the participants.

---

> ### Author Response · Authors · 2024-11-22
>
> **Foot sliding metrics** are reported as the "FS" metric in L313–L315 and all tables in the main paper.
>
> **Penetration metrics.**
>
> Our skeleton representation is based on positions and rotations exported from motion capture software. In the main paper, we visualize the predicted rotations by mapping them onto the "Xbot" model for demonstration purposes. Note that the Xbot skeleton differs from the one we use in our method, which makes penetration evaluations based on Xbot less reliable. As a result, penetration cannot be directly evaluated at the mesh level.
>
> To evaluate penetration between skeletons, we approximate the bones using triangular prisms with a distance from the centroid to a vertex = 5 cm and calculate penetration frame by frame between the two body meshes. Since boxing typically involves instantaneous contact, the proportion of frames with penetration is expected to be very low. Therefore, we report the total number of frames with penetration across the entire test set and the mean penetration volume in $cm^3$. The two-character setting results are shown below and the whole table and more details are in $\textcolor{green}{\text{Appendix I}}$.
>
> - **# Penetration** represents the number of penetration frames.
> - **Mean volume** represents the mean penetration volume ($cm^3$) over penetration frames.
>
> | Metric        | InterFormer | CVAE   | CAMDM  | T2MGPT | Ours      | GT   |
> |---------------|------------:|-------:|-------:|-------:|----------:|-----:|
> | # Penetration | 2617        | 1858   | 469    | 962    | 757       | 141  |
> | Mean Volume &#8595;| 222.83 | 218.15 | 129.14 | 365.21 | **111.84**| 21.37|
>
> From the table, we observe that our method achieves a lower mean penetration volume compared to the baselines, demonstrating its effectiveness in avoiding penetration.
>
> **The impact of sparse contact on motion generation.**
>
> To analyze the impact of sparse contact on motion generation, we first detect contact and then assess whether the agent moves backward or forward in response. Specifically, we identify contact by calculating the distance between one character’s skeleton mesh and the other’s hand joints. If the distance is less than 5 cm, we consider that the hand has successfully made contact with the other. This could include the opponent's hand hitting any part of the agent or vice versa. For these identified contact frames, we examine whether the agent’s root moves backward when the opponent's root moves forward and compute the proportion of such occurrences. For each frame, we switch the roles of "agent" and "opponent". The two-character results are shown below and the whole table is in $\textcolor{green}{\text{Appendix J}}$. A higher alignment with the ground truth indicates better performance.
>
> - **# OF** represents the number of frames that the opponent moves forward.
> - **# OF_AB** represents the number of frames that the agent moves backward when the opponent moves forward.
> - **Ratio** represents # OF_AB / # OF.
>
> | Metric     | InterFormer | CVAE   | CAMDM   | T2MGPT  | Ours     | GT     |
> |------------|------------:|-------:|--------:|--------:|---------:|-------:|
> | # OF       | 1179        | 1484   | 371     | 539     | 816      | 243    |
> | # OF_AB    | 921         | 716    | 193     | 267     | 420      | 125    |
> | Ratio &#8594; | 78.12%   | 48.25% | 52.02%  | 49.54%  |**51.47%**| 51.44% |
>
> From the table, we observe that our method achieves performance closer to the ground truth, demonstrating its ability to produce more realistic reactions.
>
> **Lacks explicit contact modeling.**
>
> Many motion generation works, such as MDM [1] and ReGenNet [2] and all of our baselines, focus on skeleton-level generation without directly addressing contact or penetration because dealing with dense contact often requires dedicated research efforts. These aspects are typically considered when addressing more intricate interaction scenarios. For studies explicitly addressing penetration or interaction within environments, works like [3] and [4] provide excellent references.
>
> We agree that contact modeling is crucial, and we will consider incorporating explicit contact modeling in our future work to enable applications in real robots. Thank you for the valuable suggestions.
>
> **Ethics concerns** have been updated in $\textcolor{green}{\text{Appendix A.3}}$.
>
> ---
> [1] Tevet, Guy, et al. "MDM: Human Motion Diffusion Model". In ICLR. 2023.
>
> [2] Xu, Liang, et al. "ReGenNet: Towards Human Action-Reaction Synthesis." In CVPR. 2024.
>
> [3] Starke, Sebastian, et al. "Neural state machine for character-scene interactions." TOG. 2019.
>
> [4] Huang, Zeyu, et al. "Spatial and Surface Correspondence Field for Interaction Transfer." In SIGGRAPH. 2024.

---

### Author Response · Authors · 2024-11-22
**General Response**

We are encouraged by the reviewers' positive feedback, noting that the problem is interesting (R1-odxr) and important (R2-T1pE), the method is sound and intuitive (R2-T1pE) but effective (R4-TLUt), efficient and versatile (R3-qtLW), the method is well-validated (R3-qtLW, R4-TLUt), and the better performance against baselines with high-quality results (R2-T1pE, R3-qtLW). The reviewers also highlighted the value of the proposed dataset (R1-odxr, R3-qtLW), and *all* the reviewers agree with the method's ability to generate long interactive movements and its potential application in VR settings. Additionally, we appreciate the recognition of our online motion decoder as a well-thought-out solution for applying VQ-VAE motion representation in autoregressive generation (R2-T1pE) and that our insight into error accumulation is interesting (R4-TLUt).

In the updated version of the paper, $\textcolor{green}{\text{changes are marked in green}}$. The following summarizes the experiments added to the paper according to the reviewers' suggestions:
- $\textcolor{green}{\text{(Appendix H)}}$ We add additional experiments on a new dataset Inter-X, proving that our method can generalize to other types of human movements (R2-T1pE, R3-qtLW, R4-TLUt).
- $\textcolor{green}{\text{(Appendix I)}}$ We add a penetration metric to evaluate the quality of the generated results (R1-odxr).
- $\textcolor{green}{\text{(Appendix J)}}$ We add a study on the impact of contact occurrences on the subsequent generated results (R1-odxr, R2-T1pE).
- $\textcolor{green}{\text{(Appendix G.3)}}$ We add an additional ablation study towards the design choice of the single-layer MLP in the diffusion network (R3-qtLW).

We appreciate the reviewers’ suggestions. For each reviewer’s detailed questions, we address their concerns directly under their respective comments.

---
### **Additional Results on the Inter-X Dataset**

Since R2-T1pE, R3-qtLW, and R4-TLUt expressed curiosity about the performance of our method on other datasets, we followed R2-T1pE’s suggestion to test our method on the Inter-X dataset and compare it with baselines.

We selected three actions from the Inter-X with varying contact frequencies: chat, kick, and dance (in increasing order of contact frequency). We use the first 22 joints from the SMPLX skeleton. The network and the training protocols remain unchanged. For the two-character FID score, we calculate each individual's per-frame, per-transition, and per-clip features, alternating between the "agent" and the "opponent". We removed the RO metric because, in actions unlike boxing, the two characters do not necessarily need always to face each other. The two-character setting results are as follows and $\textcolor{green}{\text{the whole table is in Appendix H}}$.

| "Chat"  | Per-frame FID &#8595; | Per-trans FID &#8595; | Per-clip FID &#8595; | FS &#8594;   |
|---------|----------:|----------:|---------:|-----:|
| GT      | -         | -         | -        | 0.20 |
| CAMDM   | 1.472     | 0.357     | 18.931   | 0.70 |
| T2MGPT  | 2.001     | 0.207     | 28.007   | 1.60 |
| Ours    | **1.148** | **0.160** |**15.215**|**0.31** |


| "Kick"  | Per-frame FID &#8595; | Per-trans FID &#8595; | Per-clip FID &#8595; | FS &#8594;   |
|---------|----------:|----------:|---------:|-----:|
| GT      | -         | -         | -        | 0.76 |
| CAMDM   | 1.209     | 0.604     | 19.484   | 1.99 |
| T2MGPT  | 1.770     | 0.914     | 26.250   | 1.93 |
| Ours    | **0.750** | **0.558** |**12.828**|**0.83** |

| "Dance" | Per-frame FID &#8595; | Per-trans FID &#8595; | Per-clip FID &#8595; | FS &#8594;   |
|---------|----------:|----------:|---------:|-----:|
| GT      | -         | -         | -        | 0.68 |
| CAMDM   | 2.128     | 1.192     | 34.548   | 1.21 |
| T2MGPT  | 2.714     | **0.275** | 40.943   | 1.77 |
| Ours    | **1.065** | 0.435     |**18.646**|**0.53** |

From the tables, we can observe that our method generally outperforms the selected baselines, demonstrating its ability to generalize to different types of motions effectively.

---

### Meta-Review · Area_Chair_dnpf · 2024-12-15

**Metareview:**

Scientific Claims and Findings: The paper introduces Ready-to-React, a novel approach for generating two-character online interactions.

The key contributions are:
- A novel reactive policy that generates next character poses based on past observed motions
- A model architecture incorporating diffusion head into auto-regressive model for dynamic response while mitigating error accumulation
- Demonstration of effective long-term interaction generation (>1 minute) and application to VR scenarios

Strengths:
+ Addresses an important problem in multi-agent interaction motion generation
+ Well-grounded approach combining auto-regressive model with diffusion head
+ Simple but effective architecture that achieves strong quantitative and qualitative results

Weaknesses:
- Initial experiments focused only on boxing dataset, though later validated on Inter-X
- Some methodological details were initially unclear and required clarification
- Sub-optimal reaction movements in some cases, particularly around contact/collision

The reviewers have unanimously agreed to recommend acceptance of the paper after the rebuttal period clarified their concerns.

**Additional Comments On Reviewer Discussion:**

Initial Main Comments:
- Reviewer odxr requested analysis of contact influence and physical plausibility metrics
- Reviewer T1pE noted some limitations in reaction quality and suggested broader evaluation
- Reviewer qtLW asked for clarification on architectural choices and implementation details
- Reviewer TLUt questioned novelty and significance but later revised view after discussion

Key changes from author responses include: experiments on Inter-X dataset showing generalization beyond boxing, penetration and physical plausibility metrics, ablation study on diffusion network architecture, and analysis of reaction behavior around contact events. Authors also clarified several methodological details including root representation choices and training protocols.

After these clarifications, reviewer TLUt explicitly upgraded their assessment while other reviewers maintained their support.

---

### Decision · Program_Chairs · 2025-01-22

Accept (Poster)